# Pathophysiology of Drug-Induced Hyponatremia

**DOI:** 10.3390/jcm11195810

**Published:** 2022-09-30

**Authors:** Gheun-Ho Kim

**Affiliations:** Department of Internal Medicine, Hanyang University College of Medicine, Seoul 04763, Korea; kimgh@hanyang.ac.kr; Tel.: +82-2-2290-8318

**Keywords:** aquaporin-2, kidney, nephrogenic antidiuresis, vasopressin V2 receptor, water

## Abstract

Drug-induced hyponatremia caused by renal water retention is mainly due to syndrome of inappropriate antidiuresis (SIAD). SIAD can be grouped into syndrome of inappropriate antidiuretic hormone secretion (SIADH) and nephrogenic syndrome of inappropriate antidiuresis (NSIAD). The former is characterized by uncontrolled hypersecretion of arginine vasopressin (AVP), and the latter is produced by intrarenal activation for water reabsorption and characterized by suppressed plasma AVP levels. Desmopressin is useful for the treatment of diabetes insipidus because of its selective binding to vasopressin V2 receptor (V2R), but it can induce hyponatremia when prescribed for nocturnal polyuria in older patients. Oxytocin also acts as a V2R agonist and can produce hyponatremia when used to induce labor or abortion. In current clinical practice, psychotropic agents, anticancer chemotherapeutic agents, and thiazide diuretics are the major causes of drug-induced hyponatremia. Among these, vincristine and ifosfamide were associated with sustained plasma AVP levels and are thought to cause SIADH. However, others including antipsychotics, antidepressants, anticonvulsants, cyclophosphamide, and thiazide diuretics may induce hyponatremia by intrarenal mechanisms for aquaporin-2 (AQP2) upregulation, compatible with NSIAD. In these cases, plasma AVP levels are suppressed by negative feedback. In rat inner medullary collecting duct cells, haloperidol, sertraline, carbamazepine, and cyclophosphamide upregulated V2R mRNA and increased cAMP production in the absence of vasopressin. The resultant AQP2 upregulation was blocked by a V2R antagonist tolvaptan or protein kinase A (PKA) inhibitors, suggestive of the activation of V2R-cAMP-PKA signaling. Hydrochlorothiazide can also upregulate AQP2 in the collecting duct without vasopressin, either directly or via the prostaglandin E2 pathway. In brief, nephrogenic antidiuresis, or NSIAD, is the major mechanism for drug-induced hyponatremia. The associations between pharmacogenetic variants and drug-induced hyponatremia is an area of ongoing research.

## 1. Introduction

Hyponatremia, which is defined as a serum sodium concentration < 135 mmol/L, is the most common electrolyte disorder in hospitalized patients [1]. It is often asymptomatic and found incidentally in routine laboratory tests of serum electrolytes. However, it may present with symptoms of increased intracranial pressure such as headache, nausea, and vomiting if the onset is acute or the severity of serum Na^+^ lowering is remarkable. Emergency active treatment is necessary when symptoms progress to altered consciousness including confusion, drowsiness, seizures, and coma. However, rapid correction of hyponatremia may be harmful when it is asymptomatic and chronic.

Because serum Na^+^ concentration is a function of total body sodium and water, hyponatremia can be caused by an excess of water relative to sodium in the extracellular fluid (ECF). Typically, ECF sodium depletion leads to enhanced renal water reabsorption. This is the simple mechanism of hypovolemic hyponatremia and is characterized by a low level of urine Na^+^. Relative hypovolemia, e.g., low effective circulatory volume produced by heart failure or liver cirrhosis, can also enhance renal sodium and water reabsorption and lead to hypervolemic hyponatremia. Essentially, dysnatremia is a water balance disorder caused by water excess or deficit. When the kidney mainly retains water without sodium, euvolemic hyponatremia, such as syndrome of inappropriate antidiuresis (SIAD), ensues. Old age is a common risk factor for hyponatremia because older adults do not excrete water as efficiently as those that are younger [2], probably because of the reduced glomerular filtration rate.

The term SIAD was proposed by Dr. Robertson [3] because plasma vasopressin levels were suppressed in a subgroup of patients who were diagnosed with syndrome of inappropriate antidiuretic hormone secretion (SIADH). A diagnosis of SIADH can be made when unsuppressed levels of arginine vasopressin (AVP) are detected. However, in clinical practice, ‘SIADH’ is used interchangeably with ‘SIAD’ because the clinical features are identical and accurate measurement of plasma AVP levels is clinically impractical. The reason why a subset of hyponatremic patients with the SIADH phenotype shows suppressed plasma AVP levels was elucidated partly by Feldman et al. [4]. They described two infants whose clinical and laboratory findings were consistent with SIADH but had undetectable plasma AVP levels because of gain-of-function mutations in the vasopressin V2 receptor (V2R), and coined the term ‘nephrogenic syndrome of inappropriate antidiuresis (NSIAD)’. Thus, SIAD caused by renal water retention can be classified into SIADH (with an excess of plasma AVP) and NSIAD (with appropriately suppressed plasma AVP) according to different etiologies [5].

Four major causes of SIAD are malignancies, pulmonary diseases, disorders of the central nervous system, and drugs. According to case reports, many medications are associated with hyponatremia. However, whether they all have significant cause-and-effect relationships is unclear because incidental coexistences need to be excluded in rare cases. Regarding the mechanisms of drug-induced hyponatremia, stimulation of AVP release and enhancement of AVP action in the kidney have been postulated [6]. However, the evidence for the former was limited, and the mechanisms of the latter were elusive. This paper reviews previous and recent studies and provides the current understanding of the pathophysiology of drug-induced hyponatremia. Table 1 shows the list of clinically important drugs that can induce hyponatremia [5]. Relevant mechanisms of hyponatremia induced by each category are briefly reviewed in the following sections.

## 2. Hyponatremia Induced by AVP Analogs

AVP analogs include desmopressin and oxytocin and can induce hyponatremia by acting as V2R agonists. Desmopressin selectively binds the V2R in the kidney and stimulates adenylyl cyclase activity and cAMP production in collecting duct epithelial cells [7]. This enhances osmotic water reabsorption through the upregulation of the aquaporin-2 (AQP2) water channel. Currently, desmopressin not only is used for treating diabetes insipidus but also is prescribed for relieving nocturnal polyuria in older adults. Even low doses of desmopressin can induce hyponatremia in susceptible patients with nocturnal polyuria because an advanced age is an important risk factor for hyponatremia [8]. Compared with AVP, desmopressin has a greater antidiuretic effect because of its longer half-life and selective binding to the V2R [9]. A meta-analysis reported that the incidence of desmopressin-induced hyponatremia was 7.6% in adults with nocturia [10].

Oxytocin may also induce hyponatremia when it is used in obstetrics to induce abortion and to induce or augment labor. Its antidiuretic activity is presumed by the fact that oxytocin and AVP are closely related peptides secreted from the posterior pituitary and that both are nine amino-acid peptide hormones, of which seven are identical [11]. Furthermore, the action of oxytocin as an antidiuretic hormone has been demonstrated in previous studies. Oxytocin increased osmotic water permeability in perfused inner medullary collecting ducts isolated from Sprague Dawley rats [12], and its hydro-osmotic action was mediated by V2R [13]. In Sprague Dawley rats, oxytocin treatment induced apical and basolateral translocation of the AQP2 protein along the collecting duct. This response was blocked by pretreatment with a V2R antagonist [14]. The antidiuretic action of oxytocin was also demonstrated in humans in association with AQP2 upregulation [15]. In brief, pharmacological doses of oxytocin can induce antidiuretic effects as a result of V2R stimulation and subsequent AQP2 upregulation [16].

## 3. Hyponatremia Induced by Anticancer Chemotherapeutic Agents

Hyponatremia is a common complication in cancer patients because SIAD is potentially caused by malignancies and it can be related to anticancer medical therapy as well. Vincristine, vinblastine, cisplatin, carboplatin, cyclophosphamide, and ifosfamide are the chemotherapeutic agents that are most frequently associated with hyponatremia [17]. Traditionally, these were believed to stimulate AVP release from the pituitary gland or to increase the production of AVP in the hypothalamus. Chemotherapy-induced nausea may be a potential stimulus to AVP secretion [18]. However, evidence supporting AVP hypersecretion induced by chemotherapeutic agents is limited.

Previous studies have shown that SIADH underlies the mechanism of vincristine-associated hyponatremia. A 3-year-old girl who was inadvertently administered an overdose of vincristine developed clinical features compatible with SIADH. Her blood AVP level was more than four times the normal value [19]. In addition, urinary AVP excretion was markedly elevated in a child with acute lymphatic leukemia following the administration of vincristine [20]. Furthermore, animal studies have suggested that SIADH may result from a direct toxic effect of vincristine on the neurohypophysis and the hypothalamic system [21,22].

Cisplatin is a platinum-based chemotherapeutic agent that potentially causes nephrotoxicity. It rarely induces hyponatremia via increasing plasma AVP levels [23]. Moreover, cisplatin nephrotoxicity may produce renal salt wasting causing hypovolemic hyponatremia [24]. Carboplatin may have lesser nephrotoxicity than cisplatin but is rarely associated with hyponatremia [25]. Whether plasma AVP level is increased by carboplatin administration is unclear.

Cyclophosphamide and ifosfamide are representative alkylating agents that may be associated with hyponatremia. Hyponatremia can be induced by various doses of cyclophosphamide during the treatment of malignancy and rheumatologic disease [26]. However, plasma AVP concentrations are not elevated in patients following the administration of intravenous cyclophosphamide [27,28,29]. Furthermore, antidiuresis was reported to occur in response to intravenous cyclophosphamide in patients with central diabetes insipidus [30,31], excluding the possibility of SIADH. This was confirmed by in vitro experiments using primary cultured rat inner medullary collecting duct (IMCD) cells, in which the active metabolite of cyclophosphamide (4-hydroperoxycyclophosphamide) increased cAMP production, AQP2 protein and mRNA expression, and V2R mRNA expression in the absence of vasopressin stimulation [32]. These changes were significantly ameliorated by coadministration of tolvaptan (Figure 1), suggestive of V2R-mediated NSIAD.

In contrast, elevated plasma AVP levels were found in a few cases of ifosfamide-induced hyponatremia [33,34]. Glezerman reported that ifosfamide-induced hyponatremia was corrected by the V2R antagonist conivaptan [35]. This finding supports the possibility that SIADH underlies ifosfamide-induced hyponatremia [36].

## 4. Hyponatremia Induced by Psychotropic Agents

Psychotropic agents are a broad category of drugs including antipsychotics and antidepressants used for psychiatric patients, and anticonvulsants are a category of central nervous system-acting drugs for neurologic patients. These three drug classes are the major contributors to drug-induced hyponatremia in current practice. Although they were previously described as inducing SIADH in many case reports [37], a diagnosis of SIAD is more appropriate because plasma AVP levels were undetermined [3]. More specifically, psychotropic agents were recently found to act as V2R agonists and to induce nephrogenic antidiuresis, i.e., NSIAD. In primary cultured rat IMCD cells, they stimulated V2R, increased cAMP production, and led to AQP2 upregulation in the absence of vasopressin [38]. This intrarenal mechanism is reminiscent of chlorpropamide-induced hyponatremia. Chlorpropamide is a long-acting first-generation sulfonylurea that is no longer used. It was shown to bind to the V2R within the rat renal tubular basolateral membrane in a competitive manner [39] and to increase the V2R density in rat renal papillary membranes [40]. 

### 4.1. Antipsychotic-Induced Hyponatremia

Hyponatremia is not uncommon in psychotic patients because of primary polydipsia and antipsychotic medications. According to different renal responses, the measurement of urine concentration can help distinguish between antipsychotic-induced and psychosis-induced hyponatremia [41].

Antipsychotic drugs can be grouped into first-generation antipsychotics (e.g., chlorpromazine, chlorprothixene, dixyrazine, flupentixol, fluphenazine, haloperidol, levomepromazine, melperone, perphenazine, prochlorperazine, thioridazine, and zuclopenthixol) and second-generation antipsychotics (e.g., aripiprazole, clozapine, olanzapine, paliperidone, quetiapine, risperidone, and ziprasidone). A Swedish population-based case–control study found an association between antipsychotic therapy and hospitalization due to hyponatremia. The association was stronger for first-generation antipsychotics than second-generation antipsychotics [42]. Several psychotropic drugs were reported to have the features of SIADH but without demonstrating unsuppressed plasma AVP levels [43,44,45].

On the other hand, plasma AVP levels did not change significantly when haloperidol was given to seven healthy volunteers at a dose level (1.0 mg i.m.) established to have central nervous system effects [46]. Thus, we assessed whether haloperidol can have renal action in the absence of vasopressin stimulation. Haloperidol increased cAMP production in IMCD suspensions, and this response was attenuated by co-incubation with tolvaptan. In primary IMCD cell cultures, haloperidol increased the total AQP2 protein and decreased the AQP2 phosphorylation at S261. These responses were also attenuated by co-incubation with tolvaptan or a PKA inhibitor. Immunofluorescence microscopy showed that haloperidol induced AQP2 membrane trafficking, and this change was blocked by co-incubation with tolvaptan or a PKA inhibitor. Finally, haloperidol increased V2R and AQP2 mRNA expression and CREB-1 phosphorylation at S133, and these responses were blocked by co-incubation with tolvaptan (Figure 2). We concluded that haloperidol acts as a V2R agonist in the kidney and leads to AQP2 upregulation by accelerating AQP2 transcription and AQP2 dephosphorylation at S261 [38].

### 4.2. Antidepressant-Induced Hyponatremia

Patients with depression are susceptible to hyponatremia because a variety of antidepressants have been reported to be associated with hyponatremia: tricyclic antidepressants (TCAs), monoamine oxidase inhibitors, selective serotonin reuptake inhibitors (SSRIs), serotonin-norepinephrine reuptake inhibitors (SNRIs), and mirtazapine. In addition, data from clinical and pre-clinical studies have indicated that AVP plays an important role in the pathophysiology of major depression [47]. Causality between antidepressants and hyponatremia was more frequently shown with SSRIs than with others. According to a drug surveillance program performed in Germany and Austria, SSRIs and SNRIs presented the highest risk of hyponatremia among antidepressants [48]. Of the SSRIs available, fluoxetine, paroxetine, and sertraline are the more common offenders. [49]. Older age and concomitant use of diuretics are the most important risk factors for the development of hyponatremia associated with SSRIs. Hyponatremia usually occurs within the first few weeks of drug administration, but serum sodium levels are normalized within 2 weeks of drug withdrawal [50].

It was proposed that SIADH might be derived from the effect of serotonin on 5-hydroxytryptamine receptors and the effects of norepinephrine on α1-adrenergic receptors [51]. However, in paroxetine-induced hyponatremia, no correlations between paroxetine concentration and plasma sodium or AVP levels were found [52]. Because non-suppressed plasma AVP levels were detected in only a minority of these patients, nephrogenic antidiuresis or NSIAD was suggested as the underlying mechanism of SSRI-induced hyponatremia in most patients.

The mechanism of direct renal water retention induced by SSRIs has been elucidated. Fluoxetine and sertraline are representative SSRIs that are often associated with hyponatremia [53]. An in vitro study using microperfused tubules showed that, in the absence of vasopressin, fluoxetine increased osmotic water permeability in the rat IMCD. Furthermore, AVP concentrations did not alter in rats administered with fluoxetine for 10 days, but an increased abundance of AQP2 protein was found in the kidney [54].

We investigated the regulatory pathway of AQP2 in the rat IMCD and showed that in the absence of vasopressin stimulation, sertraline upregulated AQP2 by inducing V2R-cAMP-PKA signaling. First, sertraline increased cAMP production in IMCD suspensions, and this response was attenuated by co-incubation with tolvaptan. In primary IMCD cell cultures, sertraline treatment increased total AQP2 protein and decreased phosphorylated AQP2 at S261. These responses were also attenuated by co-incubation with tolvaptan or a PKA inhibitor. Immunofluorescence microscopy showed that sertraline induced AQP2 membrane trafficking, and this change was blocked by co-incubation with tolvaptan or a PKA inhibitor. Finally, sertraline increased V2R and AQP2 mRNA expression and CREB-1 phosphorylation at S133, and these responses were blocked by co-incubation with tolvaptan (Figure 3). We concluded that sertraline acts as a V2R agonist in the kidney and leads to AQP upregulation by accelerating AQP2 transcription and AQP2 dephosphorylation at S261 [38]. The antidiuretic effect of sertraline can be applied to drug repurposing. Sertraline effectively reduced the number of wet episodes in adolescents with primary monosymptomatic enuresis [55].

### 4.3. Anticonvulsant-Induced Hyponatremia

Carbamazepine and oxcarbazepine are the anticonvulsants most commonly reported to be associated with hyponatremia in patients with epilepsy, although other anticonvulsants, such as eslicarbazepine, sodium valproate, lamotrigine, levetiracetam, and gabapentin, have also been reported to cause hyponatremia [56]. The incidence of carbamazepine-induced hyponatremia ranges widely from 4.8% to 41.5%, depending on the patient population studied [57,58,59]. As expected, the risk of hyponatremia is increased in older adults or subjects who simultaneously use other medications known to cause hyponatremia such as thiazides. It may also be increased with higher carbamazepine doses and serum carbamazepine levels and a lower initial serum sodium concentration [57,58,59].

The mechanism of anticonvulsant-associated hyponatremia has generally been considered inappropriate hypersecretion of AVP [60,61], but an experimental study has indicated a direct effect of carbamazepine on the kidney through V2R stimulation without evidence of the increased release of endogenous AVP [62]. Sekiya et al. also reported that an 18-year-old male with carbamazepine-associated hyponatremia had features of SIADH but had an undetectable level of plasma AVP and an elevated urine cyclic AMP excretion [63]. Thus, a human case of carbamazepine-induced nephrogenic antidiuresis or NSIAD was demonstrated.

It has become clear that carbamazepine has a direct action on the collecting duct V2R, leading to AQP2 upregulation. In vitro studies using microperfused tubules showed that in the absence of AVP, carbamazepine increased osmotic water absorption and AQP2 protein abundance in the rat IMCD by inducing the V2R-PKA pathway [64]. We further investigated the intracellular mechanisms of carbamazepine-induced AQP2 upregulation in the kidney. Carbamazepine increased cAMP production in IMCD suspensions, and this response was attenuated by co-incubation with tolvaptan. In primary IMCD cell cultures, carbamazepine increased the total AQP2 protein and decreased the phosphorylation of AQP2 at S261. These responses were also reversed by co-incubation with tolvaptan or a PKA inhibitor. Immunofluorescence microscopy showed that carbamazepine induced AQP2 membrane trafficking, and this change was blocked by co-incubation with tolvaptan or a PKA inhibitor. Finally, carbamazepine increased V2R and AQP2 mRNA expression and CREB-1 phosphorylation at S133, and these responses were blocked by co-incubation with tolvaptan (Figure 4). We concluded that carbamazepine acts as a V2R agonist in the kidney and leads to AQP upregulation by accelerating AQP2 transcription and AQP2 dephosphorylation at S261 [38]. Compatible with our results, carbamazepine was shown to have antidiuretic activity in seven out of nine patients with central diabetes insipidus [65].

Oxcarbazepine is a keto-analog of carbamazepine and may be more associated with hyponatremia than carbamazepine [48]. Sachdeo et al. investigated the mechanisms by which oxcarbazepine induces hyponatremia in epilepsy and healthy subjects [66]. They found that, after the water load, solute-free water clearance was reduced in both groups without a concomitant increase in the concentrations of blood AVP. Thus, oxcarbazepine-induced hyponatremia was not attributable to SIADH. It seems that oxcarbazepine and carbamazepine share common mechanisms of NSIAD, i.e., direct action on the V2R.

## 5. Thiazide-Induced Hyponatremia (TIH)

### 5.1. Clinical Presentation of TIH

Thiazide and thiazide-like diuretics are the common cause of hyponatremia that is usually induced within a few weeks of starting medication but can occur at any time and rapidly in susceptible patients. They are frequently used for the treatment of hypertension and edematous disorders. According to a retrospective cohort study, approximately 3 in 10 patients who exposed to steady use of thiazides develop hyponatremia [67]. Unlike hypokalemia, hyponatremia is dose-independent [68]. Hypertensive old women are particularly at risk of hyponatremia; the major risk factors for TIH are old age, female gender, low body mass, hypokalemia, and concurrent use of other medications that impair free water excretion [69]. Hyponatremia and inability to excrete a water load resolve within 10 to 14 days of drug withdrawal [70]. 

Serum sodium levels are variable at presentation. Mild hyponatremia, ranging from 125 to 132 mmol/L, is usually asymptomatic, although vague symptoms such as fatigue or nausea are possible [71]. More severe hyponatremia can be asymptomatic or associated with symptoms including headache, vomiting, confusion, dizziness, lethargy, seizures, and even coma. These symptoms of TIH primarily reflect osmotic water shift into brain cells rather than ECF volume depletion [72].

### 5.2. Pathogenesis of TIH

The mechanisms of TIH are complicated and not fully understood at present. Table 2 summarizes how thiazides cause hyponatremia from renal and extrarenal mechanisms. Renal mechanisms are primary and derived from the action of thiazides on renal tubules. Extrarenal mechanisms are subsidiary and include insufficient solute intake, polydipsia, and transcellular cation exchange. Low protein intake reduces urea generation and diminishes urine concentration. Patients with TIH may have a higher fluid intake at baseline and during thiazide use than normonatremic individuals [73]. Hypokalemia concurrently induced by thiazide diuretics can also promote hyponatremia. Extracellular Na^+^ will enter cells when K^+^ exits because of transcellular ion exchange. The renal mechanisms are detailed in the following paragraphs.

Thiazides inhibit the Na-Cl cotransporter (NCC) in the distal convoluted tubule, the cortical diluting segment of the nephron. Thus, urine dilution is impaired and water can be retained by thiazides [74]. Similarly, the combination of a thiazide and a K^+^-sparing diuretic such as amiloride [75,76] and spironolactone [77] can increase the risk of hyponatremia because of the enhanced urinary loss of sodium in the cortical distal tubule.

Hypovolemic hyponatremia might occur with diuretic therapy because urinary sodium loss leads to a reduction in glomerular filtration rate and enhanced reabsorption of sodium and water in the proximal tubule [78]. Hyperuricemia and low urinary uric acid excretion are characteristic findings of hypovolemia. However, patients with TIH typically show features of SIADH, including low serum uric acid concentrations (<4 mg/dL) and increased fractional excretion of uric acid (>12%) [79]. This suggests exaggerated free water reabsorption or a volume-expanded diluted state [80]. No clinical diagnostic parameters can differentiate TIH from SIAD feasibly [81]. Plasma AVP measurement in patients with TIH has produced conflicting results, with some older studies reporting elevated AVP concentrations [82,83], while more recent studies did not [73,84,85]. Ashraf et al. reported that plasma AVP was undetectable in metolazone-induced hyponatremia [86], suggestive of NSIAD. 

On the other hand, Musch and Decaux found that in diuretic-induced hyponatremia, solute depletion was the main causal factor and water retention a secondary one [87]. In seven patients with features of SIADH (e.g., serum uric acid < 4 mg/dL), an infusion of isotonic saline and potassium chloride over 3 days caused cation (Na^+^ + K^+^) retention (~600 mmoles) and increased the mean serum sodium concentration from 120 mmol/L to 133 mmol/L.

Notably, thiazide-induced renal water retention may be independent of NCC inhibition in the distal convoluted tubule. No hyponatremia is found in Gitelman syndrome or Gitelman-mimic animals carrying a loss-of-function mutation in the NCC regulator Ste20 proline-alanine-rich kinase (SPAK) [80]. Hydrochlorothiazide administration resulted in reduced urine volume in lithium-treated NCC-knockout mice [88]. In particular, thiazides may act directly on the collecting duct, where water permeability is increased by vasopressin-independent mechanisms. César and Magaldi performed in vitro microperfusion of IMCDs from AVP-deficient Brattleboro rats and showed that the addition of hydrochlorothiazide to the perfusate enhanced osmotic water permeability [89]. This effect was attenuated by adding prostaglandin E2 to the perfusate, suggesting that it involved prostaglandin signaling. We also investigated the antidiuretic mechanism of hydrochlorothiazide in rats with lithium-induced nephrogenic diabetes insipidus (NDI) and found that in association with antidiuresis, hydrochlorothiazide treatment caused a significant partial recovery of AQP2 abundance after lithium-induced downregulation [90]. 

Certain subpopulations may have a genetic predisposition to the development of TIH. In patients with TIH, hyponatremia was reproducible by single dose thiazide rechallenge where environmental factors such as sodium intake were controlled [84]. Compared with healthy older volunteers, patients with a prior history of TIH had a reduced urinary diluting ability and a greater reduction in serum osmolality [2]. These genetic associations with TIH were supported by the findings of a genetic and phenotyping analysis, suggestive of a role for genetically determined prostaglandin E2-mediated increased water permeability of the collecting ducts in the development of TIH [85]. A subgroup of patients with TIH may carry a variant allele of the prostaglandin transporter *SLCO2A1* gene that leads to a reduced ability to transport prostaglandin E2 across the apical cell membrane in the collecting duct. This reduction in prostaglandin E2 transport leads to increased luminal prostaglandin E2 and activates luminal EP4 receptors, causing membrane trafficking of AQP2 in the absence of AVP and directly enhancing urine concentration and free water absorption [91]. Consistent with this, urinary prostaglandin E2 excretion was elevated in patients with TIH who carried the *SLCO2A1* variant and returned to the control level after cessation of thiazides [85].

However, the role of thiazide diuretics in increasing urinary prostaglandin E2 excretion is not compatible with the previous notion that renal prostaglandins normally protect against TIH [2]. As mentioned above, the microperfusion study by César and Magaldi showed that the addition of prostaglandin E2 counteracts thiazide-induced water reabsorption [88]. Hydrochlorothiazide treatment in lithium-treated NCC-knockout mice reduced urinary prostaglandin E2 levels [87]. Furthermore, clinical studies report that the risk of TIH is increased by the concomitant use of nonsteroidal anti-inflammatory drugs [80,92]. Whether an inhibitor of the prostaglandin EP4 receptor can improve or prevent TIH may answer this controversial issue [93]. 

## 6. Management of Drug-Induced Hyponatremia

All potential agents that can induce hyponatremia should be discontinued. For this, meticulous history taking of previous and current medications is mandatory. Many patients are elderly and exposed to polypharmacy. Therefore, concomitant drug use should carefully be reviewed on whether multiple medications and comorbidity such as heart failure are implicated in the induction of hyponatremia. An unnecessary habit of a large fluid intake should be discouraged.

Other general treatment principles in SIAD are also applied to the management of drug-induced hyponatremia. Water restriction is the basic but weak measure to restore water balance, and small doses of furosemide can increase solute-free water clearance. Increasing oral salt and protein intake is encouraged. If hyponatremia is mild and does not accompany symptoms, discontinuation of the offending agent and restriction of water intake may be sufficient to restore serum sodium levels to normal ranges [18]. When the patients are symptomatic and acute hyponatremia is suspected, 3% hypertonic saline can be infused to elevate the serum sodium level. At this time, frequent monitoring of serum sodium levels is required to prevent overcorrection, which rarely leads to osmotic demyelination syndrome. In patients with TIH, isotonic saline infusion may be enough to slowly correct hyponatremia with or without a loop diuretic [81].

Potassium repletion will hasten the correction of hyponatremia. It is important to consider the effect of potassium supplementation on the serum sodium concentration. Because 1 mmol of retained potassium affects serum sodium as much as 1mmol of retained sodium, even partial correction of potassium depletion can cause an excessive rise in serum sodium without sodium administration [94]. This is one of the reasons why potassium depletion is a risk factor for osmotic demyelination from overcorrection of hyponatemia.

In general, re-administration of the offending agent is strongly discouraged. Thiazides should not be prescribed for patients with a history of TIH. Continued use of the potentially incriminated agents may be considered if the hyponatremia is mild, patients have responded favorably to the therapy, and alternative treatment is not available [18]. However, even mild hyponatremia may be associated with increased morbidity and mortality, particularly in the elderly, and needs likely to be avoided. Chronic mild hyponatremia increases the risk of falls because of gait disturbance and cognitive impairment [95]. Consequently, mild hyponatremia is associated with an increased risk of bone fractures. Independent of recent falls, osteoporosis and impaired bone quality may increase the fracture risk in hyponatremic patients [96]. 

Urea can induce solute diuresis, and oral urea (0.5 g/kg) is an effective and inexpensive therapy in mild chronic hyponatremia due to SIAD [97]. As described above, the major part of drug-induced hyponatremia is V2R-mediated. These drugs and selective V2R antagonists may competitively bind to the V2R. Thus, tolvaptan may be useful to treat drug-induced hyponatremia in selective patients.

## 7. Conclusions

Three drug classes (psychotropic agents, anticancer chemotherapeutic agents, and thiazide diuretics) are the major causes of drug-induced hyponatremia, which is explainable by SIAD. The final common pathway is AQP2 upregulation in the collecting duct, involving various drugs at different levels (Figure 5). Only a few drugs such as vincristine and ifosfamide were reported to stimulate AVP release (SIADH). Others can induce hyponatremia via nephrogenic antidiuresis (NSIAD). Recent in vitro studies have shown that haloperidol, sertraline, carbamazepine, and cyclophosphamide act directly on the V2R in the collecting duct and upregulate AQP2 via activation of the cAMP-PKA pathway. In the absence of AVP, these drugs can act as V2R agonists in the kidney and lead to water retention. Thiazide diuretics not only act on distal convoluted tubules to inhibit the NCC but also on collecting ducts to upregulate AQP2. These actions impair urinary dilution and enhance urinary concentration, respectively, resulting in hyponatremia. Future studies are necessary to clarify how renal prostaglandins are involved in producing TIH and whether other pharmacogenetic variants of AQP2 and its upstream modifiers are associated with drug-induced hyponatremia.

## Figures and Tables

**Figure 1 jcm-11-05810-f001:**
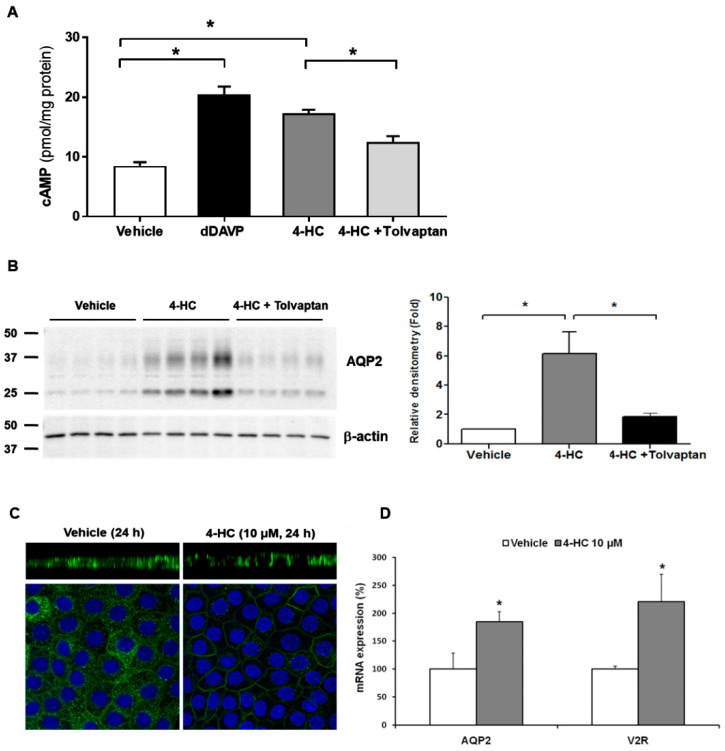
In vitro action of 4-hydroperoxycyclophosphamide (4-HC) for aquaporin-2 (AQP2) upregulation in inner medullary collecting duct (IMCD) cells. The effect of 4-HC, the active hepatic metabolite of cyclophosphamide, was tested in IMCD suspensions and primary cultured IMCD cells. (**A**) Intracellular cAMP levels were measured by ELISA in IMCD suspensions after treatment with vehicle, 10 nM dDAVP, 10 µM 4-HC, and 10 µM 4-HC + 100 nM tolvaptan at 37 °C for 30 min. (**B**) Immunoblot analysis of AQP2 was performed from primary cultured IMCD cells treated with 4-HC with and without tolvaptan. (**C**) Immunofluorescence microscopy for AQP2 in primary cultured IMCD cells shows that AQP2 targeting was induced to the plasma membrane (apical and lateral membrane) by 4-HC treatment (X-Z images). (**D**) Quantitative polymerase chain reaction data show that AQP2 and vasopressin-2 receptor (V2R) mRNA expression were increased by 4-HC treatment. Each bar represents mean ± SD. * *p* < 0.05 vs. control by the Mann–Whitney U-test. Adapted from Ref. [32] with permission.

**Figure 2 jcm-11-05810-f002:**
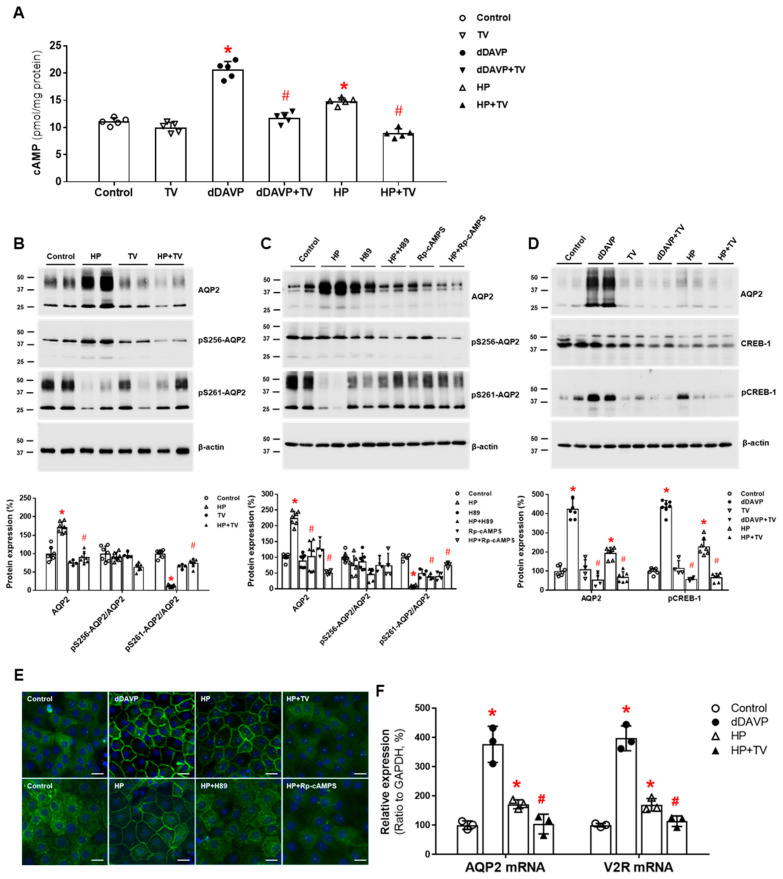
**In vitro action of haloperidol****for****aquaporin-2 (AQP2) upregulation in inner medullary collecting duct (IMCD) cells**. (**A**) cAMP production was measured by ELISA in IMCD suspensions after treatment with vehicle (control), 100 nM tolvaptan (TV), 10 nM 1-desamino-8-D-arginine vasopressin (dDAVP), and 5 μM haloperidol (HP) at 37 °C for 30 min. (**B**) Immunoblot analyses of total AQP2, pSer256-AQP2, and pSer261-AQP2 in primary cultured IMCD cells treated with 5 μM HP with and without 100 nM TV for 30 min. (**C**) Immunoblot analyses of total AQP2, pSer256-AQP2, and pSer261-AQP2 in primary cultured IMCD cells treated with 5 μM HP with and without a PKA inhibitor H89 or Rp-cAMPS. (**D**) Immunoblot analyses of total AQP2, total CREB, and pCREB-1 in primary cultured IMCD cells treated with 10 nM dDAVP and 5 μM haloperidol (HP) with and without 100 nM TV. (**E**) Immunofluorescence microscopy for AQP2 in primary cultured IMCD cells shows that AQP2 trafficking was induced by dDAVP and haloperidol (HP) but attenuated by coadministration of TV or a PKA inhibitor. (**F**) Quantitative polymerase chain reaction data show that AQP2 and vasopressin-2 receptor (V2R) mRNA expression were increased by dDAVP and HP but reversed by TV cotreatment. Each bar represents mean ± SD. * *p* < 0.05 vs. control; ^#^
*p* < 0.05 vs. HP alone by the Mann–Whitney U-test. Adapted from Ref. [38] with permission.

**Figure 3 jcm-11-05810-f003:**
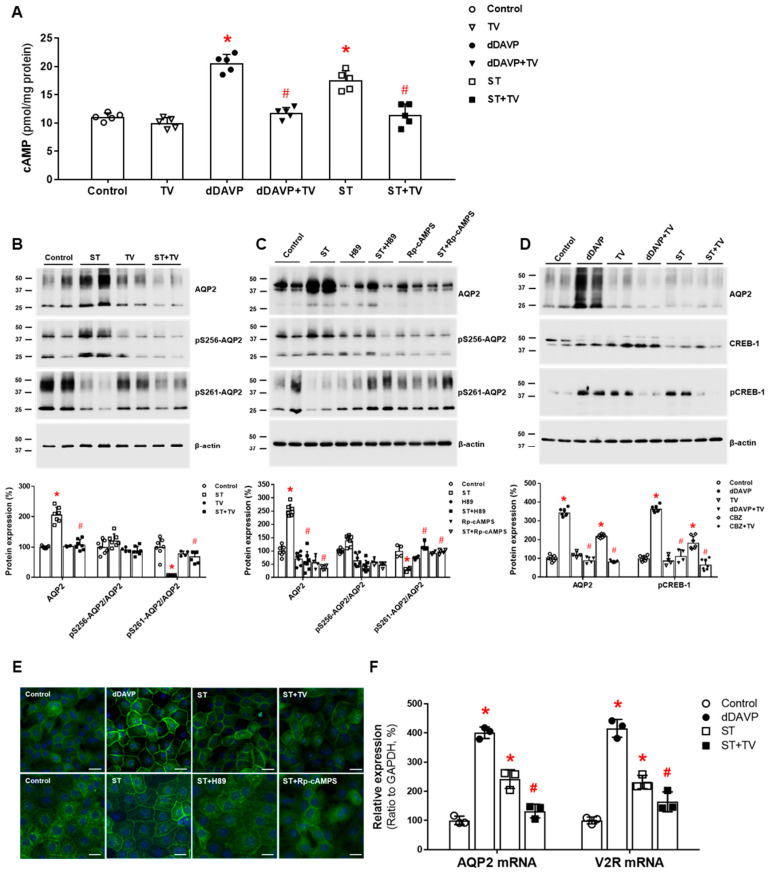
**In vitro action of sertraline for****aquaporin-2 (AQP2) upregulation in inner medullary collecting duct (IMCD) cells**. (**A**) cAMP production was measured by ELISA in IMCD suspensions after treatment with vehicle (control), 100 nM tolvaptan (TV), 10 nM 1-desamino-8-D-arginine vasopressin (dDAVP), and 1 μM sertraline (ST) at 37 °C for 30 min. (**B**) Immunoblot analyses of total AQP2, pSer256-AQP2, and pSer261-AQP2 were performed from primary cultured IMCD cells treated with 1 μM ST with and without 100 nM tolvaptan (TV) for 30 min. (**C**) Immunoblot analyses of total AQP2, pSer256-AQP2, and pSer261-AQP2 were performed from primary cultured IMCD cells treated with 1 μM ST with and without a PKA inhibitor H89 or Rp-cAMPS. (**D**) Immunoblot analyses of total AQP2, total CREB, and pCREB-1 in primary cultured IMCD cells treated with 10 nM dDAVP and 1 μM ST with and without 100 nM tolvaptan (TV). (**E**) Immunofluorescence microscopy for AQP2 in primary cultured IMCD cells shows that AQP2 trafficking was induced by dDAVP and ST but attenuated by coadministration of tolvaptan (TV) or a PKA inhibitor. (**F**) Quantitative polymerase chain reaction data show that AQP2 and vasopressin-2 receptor (V2R) mRNA expression were increased by dDAVP and sertraline (ST) but reversed by tolvaptan (TV) cotreatment. Each bar represents mean ± SD. * *p* < 0.05 vs. control; ^#^
*p* < 0.05 vs. ST alone by the Mann–Whitney U-test. Adapted from Ref. [38] with permission.

**Figure 4 jcm-11-05810-f004:**
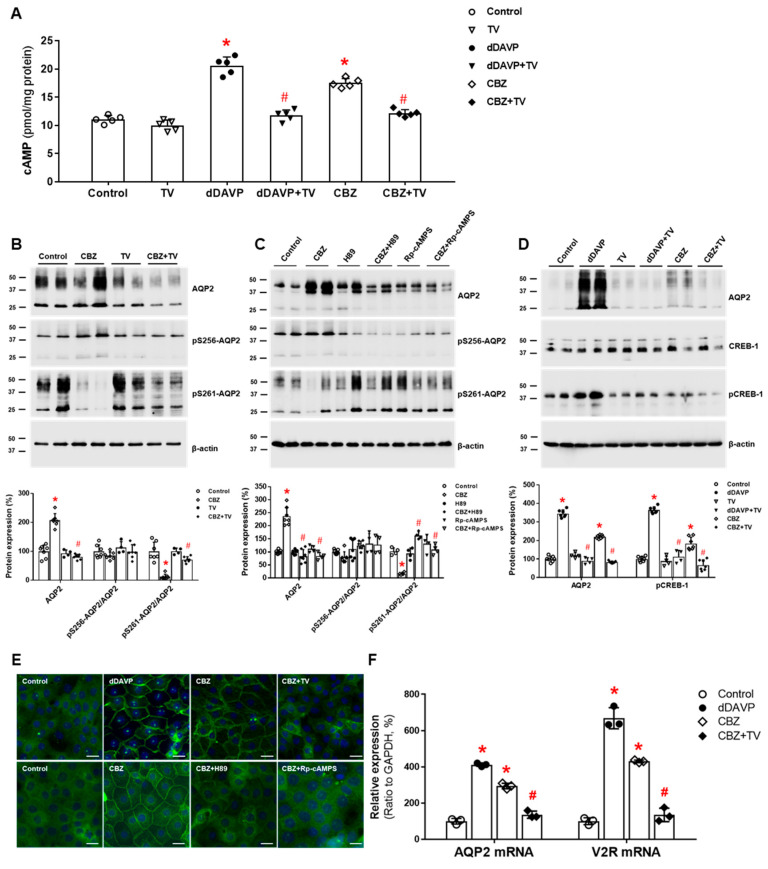
**In vitro action of carbamazepine for****aquaporin-2 (AQP2) upregulation in inner medullary collecting duct (IMCD) cells**. (**A**) cAMP production was measured by ELISA in IMCD suspensions after treatment with vehicle (control), 100 nM tolvaptan (TV), 10 nM 1-desamino-8-D-arginine vasopressin (dDAVP), or 100 μM carbamazepine (CBZ) at 37 °C for 30 min. (**B**) Immunoblot analyses of total AQP2, pSer256-AQP2, and pSer261-AQP2 in primary cultured IMCD cells treated with 100 μM CBZ with and without 100 nM tolvaptan (TV) for 30 min. (**C**) Immunoblot analyses of total AQP2, pSer256-AQP2, and pSer261-AQP2 in primary cultured IMCD cells treated with 100 μM CBZ with and without a PKA inhibitor H89 or Rp-cAMPS. (**D**) Immunoblot analyses of total AQP2, total CREB, and pCREB-1 in primary cultured IMCD cells treated with 10 nM dDAVP and 100 μM CBZ with and without 100 nM TV. (**E**) Immunofluorescence microscopy for AQP2 in primary cultured IMCD cells shows that AQP2 trafficking was induced by dDAVP and CBZ but attenuated by coadministration of TV or a PKA inhibitor. (**F**) Quantitative polymerase chain reaction data show that AQP2 and vasopressin-2 receptor (V2R) mRNA expression were increased by dDAVP and CBZ but reversed by TV cotreatment. Each bar represents mean ± SD. * *p* < 0.05 vs. control; ^#^
*p* < 0.05 vs. CBZ alone by the Mann–Whitney U-test. Adapted from Ref. [38] with permission.

**Figure 5 jcm-11-05810-f005:**
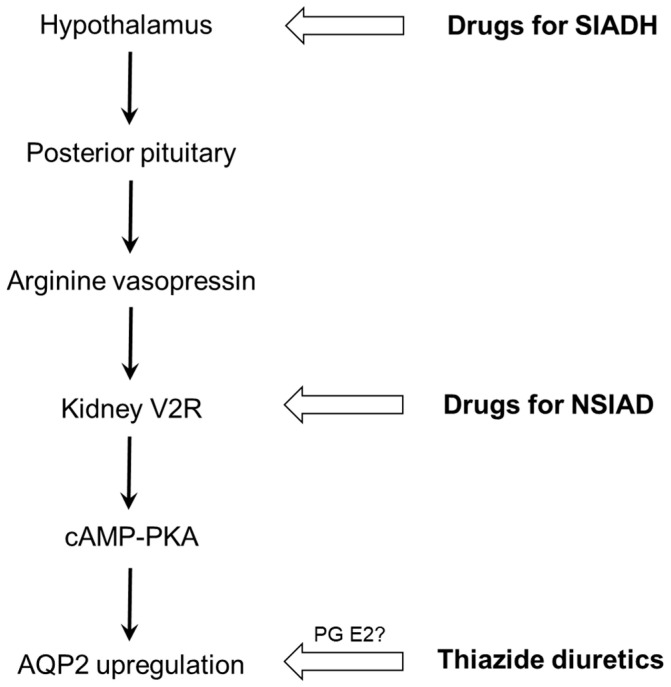
**Different levels of action for drug-induced hyponatremia.** Drugs that induce SIADH include vincristine and ifosfamide, and representative drugs for NSIAD are haloperidol, sertraline, carbamazepine, and cyclophosphamide. AQP2, aquaporin-2; cAMP, cyclic adenosine monophosphate; NSIAD, nephrogenic syndrome of inappropriate antidiuresis; PG, prostaglandin; PKA, protein kinase A; SIADH, syndrome of inappropriate antidiuretic hormone secretion.

**Table 1 jcm-11-05810-t001:** Major drugs that can cause hyponatremia.

AVP Analogs
Desmopressin (dDAVP)
Oxytocin
Drugs that stimulate the release of arginine vasopressin
Vincristine
Ifosfamide
Drugs that stimulate the vasopressin V2 receptor in the kidney
Chlorpropamide
Antidepressants: selective serotonin reuptake inhibitors
Anticonvulsants: carbamazepine
Antipsychotics: haloperidol
Cyclophosphamide
Diuretics
Thiazides: bendroflumethiazide, hydrochlorothiazide
Thiazide-like agents: chlorthalidone, indapamide, metolazone

AVP, arginine vasopressin; dDAVP, deamino D-arginine vasopressin.

**Table 2 jcm-11-05810-t002:** Mechanisms of thiazide-induced hyponatremia.

Renal (Primary)
NCC inhibition-related
Sodium loss leading to GFR reduction and enhanced proximal tubular fluid reabsorption
Impaired urinary dilution
Independent of NCC inhibition
AQP2 upregulation in the collecting duct
Direct effect
Prostaglandin E2-mediated
Extrarenal (subsidiary)
Insufficient solute intake
Excessive water intake
Coexistent hypokalemia leading to transcellular cation exchange

AQP2, aquaporin-2; GFR, glomerular filtration rate; NCC, Na-Cl cotransporter.

## Data Availability

Not applicable.

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
