# Peer review of "Pathophysiology of Drug-Induced Hyponatremia"

_jcm, 2022, doi:10.3390/jcm11195810_

Round 1
Reviewer 1 Report
Very good review on the subject
Minor comments
-Hyponatremia secondairy to thiazides eaven in those with a biology of SIADH (hypouricemia) have a large solute deficit (around 600 mmoles )(Nephron Musch W et al Nephron 2018;140(1)31)
Eaven mild hyponatremia is associated with fall,bone fracture and osteoporosis ...and need likely to be avoided
-The author must mention another easy way to treat NSIAD :oral urea witch is inexpensive and could be given during years (Europ J Internal Med 2017)
Author Response
- Hyponatremia secondairy to thiazides eaven in those with a biology of SIADH (hypouricemia) have a large solute deficit (around 600 mmoles ) (Nephron Musch W et al Nephron 2018;140(1)31)
Reply] Thank you for the comment and reference. The following sentences were added to 5.2 Pathogenesis of TIH (Page 11, bottom).
On the other hand, Musch and Decaux found that in diuretic-induced hyponatremia, solute depletion was the main causal factor and water retention a secondary one [Musch W, Decaux G. Severe Solute Depletion in Patients with Hyponatremia Due to Diuretics Despite Biochemical Pictures Similar Than Those Observed in the Syndrome of Inappropriate Secretion of Antidiuretic Hormone. Nephron. 2018;140:31-38]. In seven patients with features of SIADH (e.g., serum uric acid < 4 mg/dL), an infusion of isotonic saline and potassium chloride over 3 days caused cation (Na+ + K+) retention (~ 600 mmoles) and increased the mean serum sodium concentration from 120 mmol/L to 133 mmol/L.
- Eaven mild hyponatremia is associated with fall, bone fracture and osteoporosis ...and need likely to be avoided
Reply] Thank you for the comment. The following sentences were added to 6. Management of drug-induced hyponatremia (Page 13, the second last paragraph).
However, even mild hyponatremia may be associated with increased morbidity and mortality particularly in the elderly and needs likely to be avoided. Chronic mild hyponatremia increases the risk of falls because of gait disturbance and cognitive impairment [Renneboog B, Musch W, Vandemergel X, Manto MU, Decaux G. Mild chronic hyponatremia is associated with falls, unsteadiness, and attention deficits. Am J Med. 2006;119(1):71.e1-8]. Consequently, mild hyponatremia is associated with an increased risk of bone fractures. Independent of recent falls, osteoporosis and impaired bone quality may increase the fracture risk in hyponatremic patients [Hoorn EJ, Rivadeneira F, van Meurs JB, Ziere G, Stricker BH, Hofman A, Pols HA, Zietse R, Uitterlinden AG, Zillikens MC. Mild hyponatremia as a risk factor for fractures: the Rotterdam Study. J Bone Miner Res. 2011;26(8):1822-8].
- The author must mention another easy way to treat NSIAD: oral urea witch is inexpensive and could be given during years (Europ J Internal Med 2017)
Reply] Thank you for the comment and reference. The following sentences were added to 6. Management of drug-induced hyponatremia (Page 13, bottom).
Urea can induce solute diuresis, and oral urea (0.5 g/kg) is an effective and inexpensive therapy in mild chronic hyponatremia due to SIAD [Decaux G, Gankam Kengne F, Couturier B, Musch W, Soupart A, Vandergheynst F. Mild water restriction with or without urea for the longterm treatment of syndrome of inappropriate antidiuretic hormone secretion (SIADH): Can urine osmolality help the choice? Eur J Intern Med. 2018;48:89-93].
Reviewer 2 Report
Thank you for giving me the opportunity to review the article. The author nicely reviewed the pathophysiology of drug-induced hyponatremia.
1: In line 462, potassium supplementation is stated to speed improvement of hyponatremia, but this leads rather to overcorrection of hyponatremia. It is necessary to state that when both hypokalemia and hyponatremia are present, great care should be taken in potassium supplementation.
Author Response
In line 462, potassium supplementation is stated to speed improvement of hyponatremia, but this leads rather to overcorrection of hyponatremia. It is necessary to state that when both hypokalemia and hyponatremia are present, great care should be taken in potassium supplementation.
Reply] Thank you for the comment. The following sentences were added to 6. Management of drug-induced hyponatremia (Page 13, middle).
It is important to consider the effect of potassium supplementation on the serum sodium concentration. Because 1 mmol of retained potassium affects serum sodium as much as 1mmol of retained sodium, even partial correction of potassium depletion can cause an excessive rise in serum sodium without sodium administration [Adrogué HJ, Madias NE. The challenge of hyponatremia. J Am Soc Nephrol. 2012;23(7):1140-8]. This is one of the reasons why potassium depletion is a risk factor for osmotic demyelination from overcorrection of hyponatemia.
Thank you very much.